# CryoCCD: Conditional Cycle-consistent Diffusion with Biophysical Modeling for Cryo-EM Synthesis

## Abstract

Single-particle cryo-electron microscopy (cryo-EM) has become a cornerstone of structural biology, enabling near-atomic resolution analysis of macromolecules through advanced computational methods. However, the development of cryo-EM processing tools is constrained by the scarcity of high-quality annotated datasets. Synthetic data generation offers a promising alternative, but existing approaches lack thorough biophysical modeling of heterogeneity and fail to reproduce the complex noise observed in real imaging. To address these limitations, we present CryoCCD, a synthesis framework that unifies versatile biophysical modeling with the first conditional cycle-consistent diffusion model tailored for cryo-EM. The biophysical engine provides multi-functional generation capabilities to capture authentic biological organization, and the diffusion model is enhanced with cycle consistency and mask-guided contrastive learning to ensure realistic noise while preserving structural fidelity. Extensive experiments demonstrate that CryoCCD generates structurally faithful micrographs, enhances particle picking and pose estimation, as well as achieves superior performance over state-of-the-art baselines, while also generalizing effectively to held-out protein families.

## 1 Introduction

In recent years, single-particle cryo-electron microscopy (cryo-EM) has emerged as an essential technique in structural biology, enabling near-atomic resolution reconstructions of macromolecules in their native states (Cheng, 2015; Kühlbrandt, 2014). By vitrifying biological specimens and imaging them with high-energy electron beams, cryo-EM has significantly deepened our understanding of protein architectures and molecular mechanisms, thereby accelerating drug discovery and expanding biological insight (Nogales, 2016; Lyumkis, 2019). However, its full potential is constrained by several data-centric limitations, including the scarcity of datasets, extremely low signal-to-noise ratios (SNR), and reliance on labor-intensive annotation processes (Nakane et al., 2020; Henderson, 2013). These challenges collectively impede the development of robust models for downstream tasks such as particle picking (Bepler et al., 2019), pose estimation (Levy et al., 2022), and 3D reconstruction (Zhong et al., 2021a;b).

To support the development of learning-based cryo-EM algorithms (Zhu et al., 2017b; Gupta et al., 2021; Zhong et al., 2021a; Wu et al., 2021; Jiang et al., 2025), recent efforts have focused on generating synthetic data (Rullgård et al., 2011; Vulović et al., 2013; Dhakal et al., 2023) through biophysically inspired modeling. Several cryo-EM–specific frameworks, including InsilicoTEM (Vulović et al., 2013) and LBPN (Kiewisz et al., 2025), simulate high-fidelity multi-angle projection processes to approximate realistic imaging conditions. While effective, these methods often incur substantial computational costs. VirtualIce (Noble et al., 2023) improves efficiency by simulating particle behaviors such as aggregation, overlap, and preferred orientations on top of real vitrified backgrounds. However, it relies on simplified Gaussian noise injection, which fails to capture the complex and structured noise patterns observed in real cryo-EM micrographs (Li et al., 2022).

Despite these advances, existing biophysical simulation approaches (Vulović et al., 2013; Rullgård et al., 2011) still face two major limitations. First, they offer limited support for modeling structural diversity and spatial variability, essential for generating datasets tailored to specific biological con-

texts (Zhong et al., 2021a; Liu et al., 2023). Second, they typically assume additive Gaussian noise, whereas real cryo-EM data contain a mixture of detector noise, electron scattering artifacts, radiation damage, and heterogeneous background signals (Li et al., 2022; Parkhurst et al., 2024). To mitigate this gap, CryoGEM (Zhang et al., 2024) introduces a physics-informed generative framework that better models noise distributions. Nevertheless, its GAN-based formulation lacks bidirectional constraints (Zhu et al., 2017a), making it prone to structure distortion and incapable of controllable noise synthesis.

To address these limitations, we propose **CryoCCD**, **C**onditional **C**ycle-consistent **D**iffusion with Biophysical Modeling for **Cryo**-EM Synthesis. We employ a modular biophysical engine for multi-scale cryo-EM synthesis, using structurally diverse particles from the PDB and AlphaFold3 (ww-PDB consortium, 2018; Abramson et al., 2024) to simulate virtual cellular specimens and modeling imaging physics to generate micrographs that faithfully capture heterogeneity. For realistic noise simulation, CryoCCD is the first to apply diffusion models to cryo-EM data synthesis. Compared to GAN-based methods (Zhang et al., 2024; Harar et al., 2025), which are prone to mode collapse, conditional diffusion models provide more stable generation. To preserve structural fidelity, we introduce a cycle-consistent approach that aligns positional information during domain translation. Furthermore, mask-guided contrastive learning enhances the representation of fine-grained features such as edges, textures, and spatial noise patterns.

We validate CryoCCD on six diverse cryo-EM datasets, showing consistently superior scores on both FID and CMMD compared to traditional noise models and recent generative baselines. Beyond visual realism, our framework improves downstream tasks: particle picking achieves higher AUPRC, pose estimation attains lower angular error, and both tasks generalize robustly to unseen protein families, highlighting CryoCCD's strong transferability. The simulator codebase will be open-sourced to facilitate further research.

Our contributions are summarized as follows:

- We develop a comprehensive modeling engine that encodes biophysical priors, enabling multi-functional cryo-EM synthesis with compositional and conformational heterogeneity.

- We introduce the first diffusion framework for cryo-EM noise generation, where cycle consistency preserves structural integrity and mask-guided contrastive learning enhances fine-grained representation, producing noise that is more realistic than traditional, GAN-based, and diffusion-based approaches.

- We validate CryoCCD on six public cryo-EM datasets, achieving state-of-the-art performance on standard metrics (FID, CMMD), significant improvements in particle picking and pose estimation, and strong generalization to unseen protein families.

## 2 RELATED WORK

**Cryo-EM/ET Synthesis.** The development of cryo-EM/ET data synthesis has evolved from early physics-based models of electron scattering (Cowley and Moodie, 1957; Vulović et al., 2013) to more realistic frameworks that account for imaging noise and sample heterogeneity (Himes and Grigorieff, 2021; Zhang et al., 2020; Dsouza et al., 2023; Zheng et al., 2023; Joosten et al., 2024). To balance realism and controllability, hybrid simulators like VirtualIce (Noble et al., 2023) and cryo-TomoSim (Purnell et al., 2023) generate annotation-ready data with physical priors. More recently, specialized approaches have been developed to handle cryo-ET challenges, such as Pol-Net (Martinez-Sanchez et al., 2024) for cellular variability and LBPN (Kiewisz et al., 2025) for the missing wedge problem. However, most methods simulate noise by adding Gaussian perturbations, which fail to capture the complex noise patterns observed in real micrographs. To bridge this gap, CryoETGAN (Wu et al., 2022) introduces cycle-consistent unpaired translation, while FakET (Harar et al., 2025) leverages neural style transfer for efficient synthesis. CryoGEM (Zhang et al., 2024) further advances realism by integrating physics-based simulation with mask-guided contrastive learning. In this work, we aim to improve the cryo-EM synthesis method by introducing biophysical modeling for structural diversity and diffusion models for realistic noise.

**Unpaired Image-to-Image Translation.** Realistic noise generation in cryo-EM can be cast as an unpaired image-to-image translation task, where clean simulated micrographs are mapped to

realistic noisy ones without paired supervision. In vision, this has been widely studied, from early GAN-based frameworks (Zhu et al., 2017a; Yi et al., 2017; Lee et al., 2018; Royer et al., 2020; Choi et al., 2018) to variants with shared latent spaces (Liu et al., 2017), attention (Kim et al., 2020), and contrastive learning (Park et al., 2020; Jung et al., 2022; Wang et al., 2021; Zheng et al., 2021). Extending these ideas, GAN-based models such as CryoETGAN (Wu et al., 2022) and CryoGEM (Zhang et al., 2024) have been applied to cryo-EM/ET, though often limited by instability and mode collapse. In contrast, our method enables stable and structurally faithful noise generation.

**Diffusion Models.** Diffusion models have emerged as a powerful generative framework, with DDPMs defining a forward noise process and learning its reversal (Ho et al., 2020; Song et al., 2022). Early efforts focused on sampling efficiency and image quality (Nichol and Dhariwal, 2021; Dhariwal and Nichol, 2021; Rombach et al., 2022), while recent advances enable conditional generation via classifier guidance and context-aware conditioning (Ho and Salimans, 2022; Yang et al., 2024; Zhang et al., 2023). For unpaired image-to-image translation, diffusion models offer greater stability and diversity than GANs. CycleDiffusion (Wu and De la Torre, 2022) introduced a unified latent space to enforce cycle consistency, improving fidelity. Currently, CycleNet (Xu et al., 2023) introduces a lightweight cycle consistency regularizer for text-guided diffusion. Despite these advances, diffusion models remain underexplored in cryo-EM synthesis. To fill this gap, we introduce the first conditional, cycle-consistent diffusion framework tailored for this domain.

## 3 BIOPHYSICAL MODELING FOR DIVERSE STRUCTURAL SIMULATION

We introduce a unified biophysical modeling framework that leverages both experimental and generated structures, and scale-adaptive placement to synthesize high-fidelity, biologically grounded cryo-EM micrographs. By integrating data-driven and probabilistic placement strategies with realistic imaging physics, our pipeline captures the compositional and spatial heterogeneity of cellular specimens. Full specifications are provided in Appendix A.

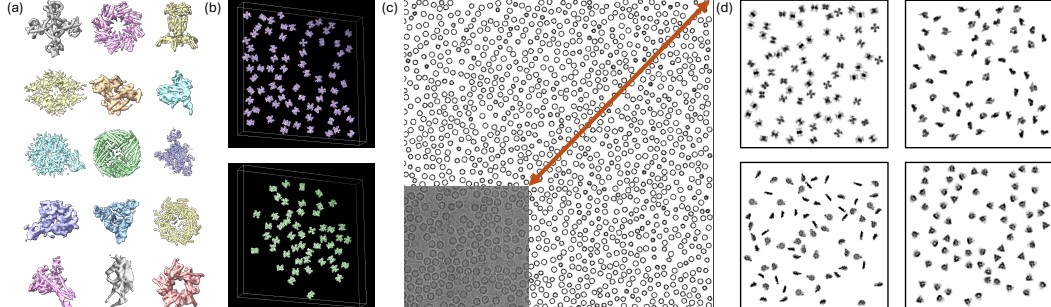

Figure 1: Cryo-EM simulation results: (a) Visualization of the processed particles. (b) Different placement strategies are adopted based on the particles' properties. (c) Generation of multi-scale synthetic data from real cryo-EM images according to EMPIAR-10421. (d) Visualization of simulated cryo-EM images.

**Library Construction.** To enrich the structural basis for simulation, we construct a heterogeneity-supporting model library by combining a broad collection of experimentally resolved PDB structures, spanning molecular weights from small enzymes (~50 kDa) to large viral capsids (>50,000 kDa) and symmetry classes from high-order tetrahedral assemblies to asymmetric $C_1$ complexes, and combining AlphaFold predictions on curated proteomes to fill gaps in experimental structures (see Figure 5), thereby expanding structural coverage to improve the robustness and generalizability. All models are then standardized through Gaussian smoothing and isosurface extraction, followed by triangulated mesh conversion while preserving fine-scale features. The processed particles are shown in Figure 1(a).

**Virtual Sample Modeling.** To assemble in-silico specimens for imaging, we build virtual samples that balance experimental fidelity and controllable diversity. The steps are as follows:

- **Particle Placement.** To achieve biologically plausible positioning and orientation of macromolecules, we combine empirical annotations with probabilistic sampling under tunable fidelity-diversity tradeoffs. RELION-derived (Kimanius et al., 2021) picks and angles are mapped from pixel position to volume coordinates to yield translations $T_{\exp}$ and quaternion orientations $q_{\exp}$. To augment sampling beyond mapped particles, we sample translations $T_{\text{syn}}$ from density-matched or uniform distributions calibrated to empirical radial functions and orientations $q_{\text{syn}}$ uniformly on $S^3$. Confidence-weighted blending of $(T_{\exp}, q_{\exp})$ and $(T_{\text{syn}}, q_{\text{syn}})$ ensures fidelity and diversity.

- **Class-Specific Distribution Modeling.** To recapitulate distinct spatial arrangements of molecular subtypes, we apply particle-type–specific priors inferred from experimental distributions (Figure 1(b)). A dedicated module applies distribution rules extracted from experimental data: soluble enzymes disperse uniformly in the volume, ribosomes cluster according to measured inter-particle distances to mimic polysomes, and viral capsids follow separation distributions observed on cryo-EM grids to avoid clashes.

- **Multi-Scale Volume Modeling.** To enable simulation across multiple scales, our simulator adjusts mesh complexity and placement parameters across scales (Figure 1 (c)). At larger scales, simplified meshes and broader sampling capture global spatial patterns. While at finer scales, detailed meshes and denser sampling with stricter collision checks preserve sub-nanometer features. This scale-adaptive modeling ensures consistent biological realism from whole-cell panoramas down to molecular interfaces.

- **Ice-Layer Modeling.** To reflect imaging conditions shaped by vitrified ice, we simulate an ice layer with realistic topography and density fluctuations. We generate a vitreous ice slab with thickness drawn from a log-normal distribution and modulate its surface via Perlin-noise to introduce realistic thickness variations without real-sample input. The ice density field incorporates Gaussian fluctuations to simulate beam-induced noise before particle embedding.

**Projection and Electrostatic Potential Assembly.** To produce high-fidelity micrographs from the virtual sample, we compute the total electrostatic potential from all embedded components as

$$\Phi(\mathbf{r}) = \sum_{i=1}^{L} \rho_i\big(R(q_i)^{-1}(\mathbf{r} - T_i)\big), \tag{1}$$

and project it under the weak-phase approximation:

$$P(x,y) = \int_{-z_0/2}^{z_0/2} \Phi(x,y,z)\, \mathrm{d}z. \tag{2}$$

The generated projections are shown in Figure 1 (d). We then apply the instrument contrast transfer function in Fourier space and perform an inverse Fourier transform to obtain a noise-free micrograph for subsequent diffusion-based noise synthesis (see Appendix A.1).

## 4 CONDITIONAL CYCLE-CONSISTENT DIFFUSION FOR REALISTIC NOISE TRANSLATION

Realistic noise generation in cryo-EM can be formulated as an unpaired image-to-image translation problem, where noise-free simulated micrographs are translated into realistic noisy ones without requiring paired supervision. Unlike traditional Gaussian or Poisson noise injection, which fails to capture structured background and detector-specific artifacts, this formulation allows learning complex noise distributions directly from real cryo-EM data. However, existing GAN-based approaches (Zhang et al., 2024) often suffer from mode collapse and lack explicit constraints to preserve structural fidelity. To overcome these limitations, we introduce CryoCCD, which leverages the stability of diffusion models, enforces cycle-consistency to maintain structural integrity, and incorporates mask-guided conditioning for fine-grained noise modeling.

### 4.1 CONDITIONAL CYCLE-CONSISTENT DIFFUSION

We first define two generative diffusion models: $G_{AB}$, which translates synthetic micrographs into the realistic cryo-EM domain, and $G_{BA}$, which reconstructs synthetic-like images from real micrographs. This bidirectional mapping enables unpaired training while preserving structural fidelity.

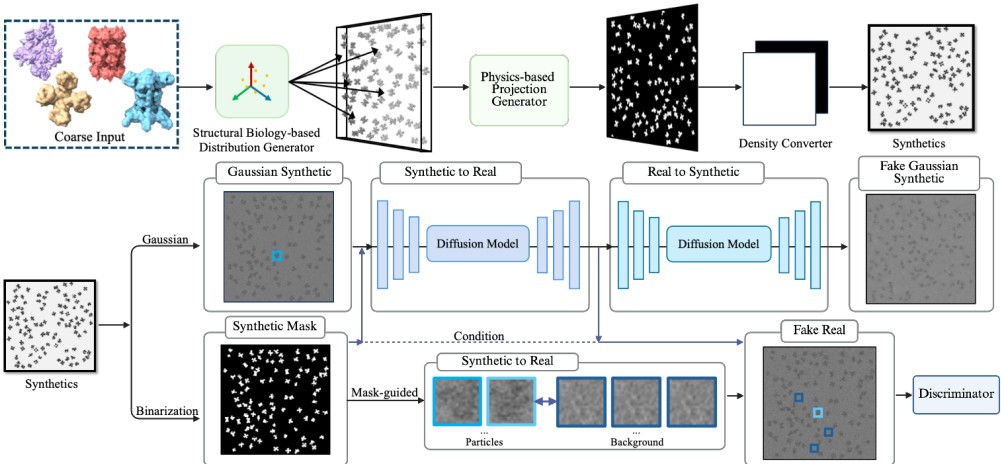

Figure 2: CryoCCD pipeline: (1) The input structures are inserted into a volume through specific placement strategies, which then undergoes physics-based projection and density conversion to generate synthetic images. (2) In the image translation, we use two diffusion models, mask-guided contrastive learning, and discriminator to achieve realistic synthetic-to-real image translation.

The forward diffusion process over $T$ timesteps follows:

$$q(\mathbf{x}_t \mid \mathbf{x}_{t-1}) = \mathcal{N}(\mathbf{x}_t; \sqrt{1 - \beta_t}\mathbf{x}_{t-1}, \beta_t\mathbf{I}), \tag{3}$$

where $\{\beta_t\}_{t=1}^T$ is the noise schedule. The reverse process is:

$$p_\theta(\mathbf{x}_{t-1} \mid \mathbf{x}_t, \mathbf{m}) = \mathcal{N}(\mathbf{x}_{t-1}; \mu_\theta(\mathbf{x}_t, t, \mathbf{m}), \sigma_t^2\mathbf{I}), \tag{4}$$

with mean function:

$$\mu_\theta(\mathbf{x}_t, t, \mathbf{m}) = \frac{1}{\sqrt{\alpha_t}} \left( \mathbf{x}_t - \frac{1 - \alpha_t}{\sqrt{1 - \bar{\alpha}_t}} \epsilon_\theta(\mathbf{x}_t, t, \mathbf{m}) \right), \tag{5}$$

where $\alpha_t = 1 - \beta_t$, $\bar{\alpha}_t = \prod_{s=1}^t \alpha_s$, and $\epsilon_\theta$ is the noise prediction network.

The denoising objective minimizes the expected squared error over timesteps $t$, clean images $\mathbf{x}_0$, noise $\epsilon$, and masks $\mathbf{m}$:

$$\mathcal{L}_{\text{diff}} = \mathbb{E}_{t,\mathbf{x}_0,\epsilon,\mathbf{m}} \left[ \|\epsilon - \epsilon_\theta(\mathbf{x}_t, t, \mathbf{m})\|_2^2 \right]. \tag{6}$$

The segmentation mask $\mathbf{m}$ guides the model to attend to particle regions, enhancing structural preservation. Sampling is accelerated using conditional sampler UniPC (Zhao et al., 2023) guided by mask-based conditioning:

$$\tilde{\mathbf{x}}_{t_i} = \frac{\alpha_{t_i}}{\alpha_{t_{i-1}}} \tilde{\mathbf{x}}_{t_{i-1}} - \sigma_{t_i}(e^{h_i} - 1)\epsilon_\theta(\tilde{\mathbf{x}}_{t_{i-1}}, t_{i-1}, \mathbf{m}) - \sigma_{t_i} B(h_i) \sum_{k=1}^p \frac{a_k}{r_k} D_k, \tag{7}$$

where $h_i = \lambda_{t_i} - \lambda_{t_{i-1}}$ is the step size in half log-SNR domain, and

$$D_k = \epsilon_\theta(\tilde{\mathbf{x}}_{s_k}, s_k, \mathbf{m}) - \epsilon_\theta(\tilde{\mathbf{x}}_{t_{i-1}}, t_{i-1}, \mathbf{m}) \tag{8}$$

with auxiliary timesteps $s_k = t_\lambda(r_k h_i + \lambda_{t_{i-1}})$. Here $B(h_i) = O(h_i)$ and $\{a_k\}_{k=1}^p$ are coefficients for $(p + 1)$-th order accuracy.

To ensure consistency across domains without paired supervision, we impose a cycle-consistency constraint using the $L_1$ norm:

$$\mathcal{L}_{\text{cyc}} = \mathbb{E}_{\mathbf{x}\sim p_{\mathcal{A}}} \left[\|G_{BA}(G_{AB}(\mathbf{x}, \mathbf{m}), \mathbf{m}) - \mathbf{x}\|_1\right] + \mathbb{E}_{\mathbf{y}\sim p_{\mathcal{B}}} \left[\|G_{AB}(G_{BA}(\mathbf{y}, \mathbf{m}), \mathbf{m}) - \mathbf{y}\|_1\right], \tag{9}$$

where $p_{\mathcal{A}}$ and $p_{\mathcal{B}}$ are the data distributions over domains $\mathcal{A}$ and $\mathcal{B}$ respectively.

## 4.2 MASK-GUIDED CONTRASTIVE LEARNING

Standard contrastive sampling often conflates particle and background regions, weakening structural discrimination under high noise. To overcome this limitation, we introduce mask-guided contrastive learning, where segmentation masks explicitly define particles as positives and background as negatives, improving fine-grained feature representation in edges, textures, and spatial noise patterns. The contrastive loss is defined as:

$$\mathcal{L}_{\text{NCE}} = -\log \frac{\exp\left(\cos(q, k^+)/\tau\right)}{\exp\left(\cos(q, k^+)/\tau\right) + \sum_{k^-} \exp\left(\cos(q, k^-)/\tau\right)}, \tag{10}$$

where $q$ is the query feature, $k^+$ is its corresponding positive, and $k^-$ are sampled negatives. Here $\cos(\cdot, \cdot)$ denotes cosine similarity, and $\tau$ is a temperature parameter.

To further improve local realism, we incorporate a PatchGAN (Demir and Unal, 2018) discriminator $D_B$, trained with the following adversarial loss:

$$\mathcal{L}_{\text{GAN}} = \mathbb{E}_{\mathbf{y} \sim \mathcal{B}}[\log D_B(\mathbf{y})] + \mathbb{E}_{\mathbf{x} \sim \mathcal{A}}[\log(1 - D_B(G_{AB}(\mathbf{x}, \mathbf{m})))]. \tag{11}$$

We also apply physics-informed preprocessing to synthetic inputs, including weight-map normalization $\tilde{\mathbf{x}} = \text{IN}(w \odot \mathbf{x})$ and CTF-based noise simulation, to better match the statistical characteristics of real cryo-EM images.

The overall training objective combines all components where $\lambda$ are empirically tuned weights:

$$\mathcal{L} = \mathcal{L}_{\text{diff}} + \lambda_{\text{GAN}}\mathcal{L}_{\text{GAN}} + \lambda_{\text{cyc}}\mathcal{L}_{\text{cyc}} + \lambda_{\text{NCE}}\mathcal{L}_{\text{NCE}}, \tag{12}$$

## 5 EXPERIMENTS

**Datasets.** We trained CryoCCD using synthetic particles derived from 15 publicly available EMPIAR datasets (Iudin et al., 2023) and the details are provided in Appendix B.1. For visual quality evaluation purposes, we selected *six* datasets: 1) **TRPV1** (Liao et al., 2013), a tetrameric membrane channel resolved at 3.4 Å; 2) $\beta$-**galactosidase** (Bartesaghi et al., 2015), a 2.2 Å map with ∼800 solvent peaks; 3) **Rhino/enterovirus** (Abdelnabi et al., 2019), small icosahedral virions requiring symmetry-expansion and high-fidelity difference mapping to resolve the conserved VP1-VP3 pocket; 4) **Innexin-6** (Burendei et al., 2020), nanodisc-embedded hemichannels where lipid-induced pore closure demands focused classification and membrane-signal subtraction; 5) **MLA complex** (Mann et al., 2021), apo/ATP/ADP states that necessitate heterogeneous 3-D clustering and multi-body refinement to trace lipid transport; and 6) **GroEL** (Godek et al., 2024), a D7-symmetry chaperonin. Comprehensive descriptions of all datasets are provided in Appendix B.1.

**Baselines.** We evaluate the performance of our method compared to several traditional noise baselines and deep generative models. We use **Poisson** noise, **Gaussian** noise, and Poisson-Gaussian mixed noise (**Poi-Gau**) as traditional baselines. We choose **CryoGEM** (Zhang et al., 2024), **CycleDiffusion** (Wu and De la Torre, 2022) and **CycleNet** (Xu et al., 2023) as deep generative baselines. In Appendix B.2, we detail their specific settings.

**Implementation Details.** All experiments are conducted on a NVIDIA GeForce RTX A100 GPU and trained for 50 epochs. The training dataset consists of 500 synthetic and 500 real images, with all images standardized to a resolution of 1024×1024 pixels. Further implementation details of sampler setups are provided in Appendix B.3.

### 5.1 VISUAL QUALITY

Visualization examples of each step in the CryoCCD pipeline are shown in Figure 3 (more visualizations in Appendix A.1). We compared CryoCCD against various baselines through visual assessment of generated images of evaluation datasets. Figure 4 illustrates that our method produces significantly more authentic micrographs with superior noise characteristics while preserving structural details such as particle boundaries, internal textures, and contrast between particles and background. Conventional approaches (Gaussian, Poisson, and Poisson-Gaussian mixed noise models) consistently failed to capture the complex noise patterns inherent in real cryo-EM data, often

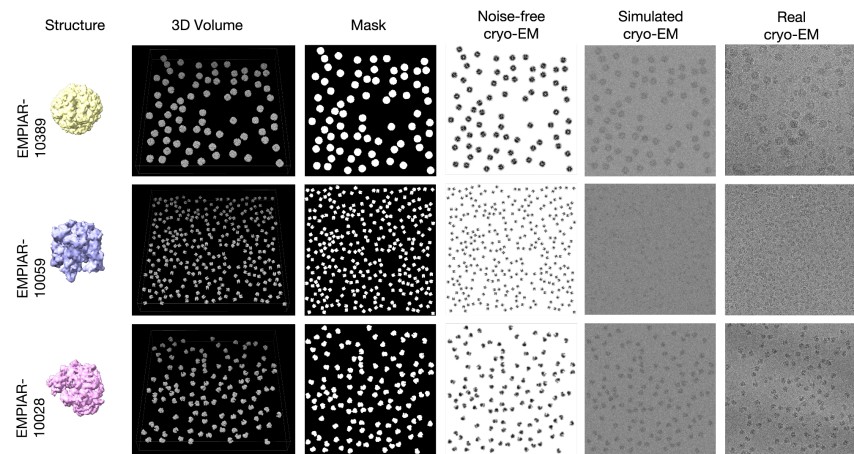

Figure 3: Visual Examples of CryoCCD Pipeline.

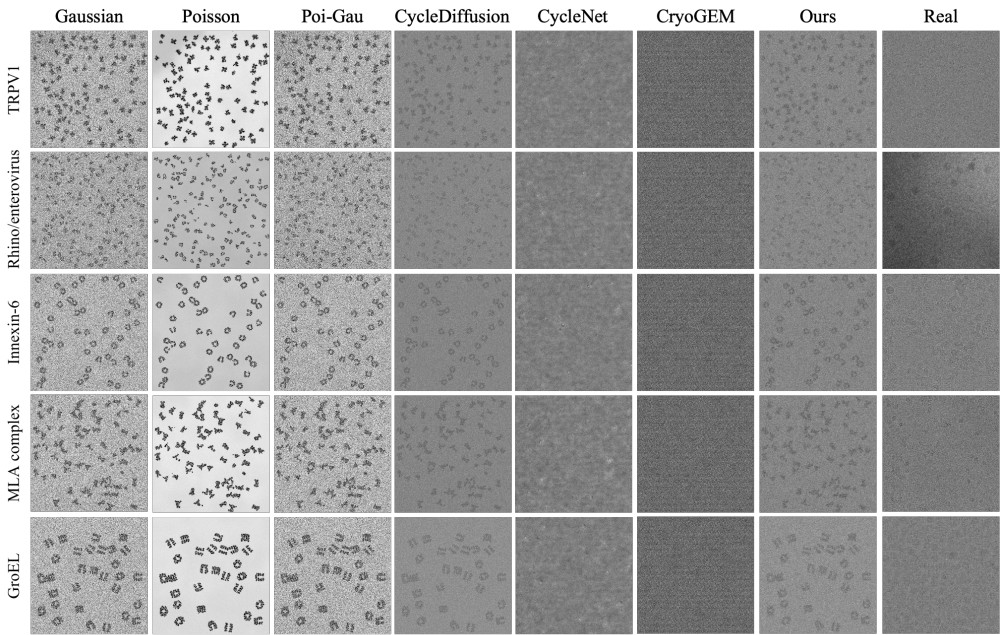

Figure 4: Comparison between real images and generated fake real images. Our method produces more authentic micrographs with superior noise characteristics across all datasets.

resulting in unnatural global intensity distributions and oversmoothed textures. Among deep generative models, CycleDiffusion often produced images where the particle boundaries were blurred and indistinguishable from the background. CryoGEM and CycleNet tended to collapse when handling our composite synthetic datasets, failing to generate recognizable particles.

Quantitatively, we report both Fréchet Inception Distance (**FID**) and Class-Mean Maximum Discrepancy with learned features (**CMMD**) (Jayasumana et al., 2024) to complement FID; Table 1 compares the two metrics across six cryo-EM datasets and shows that **CryoCCD** attains the lowest scores on both against Gaussian/Poisson/Poi-Gau, recent GAN-based baseline (CryoGEM), and diffusion-based baseline (CycleDiffusion, CycleNet).

## 5.2 PARTICLE PICKING

To validate the realism of the noise characteristics, we evaluated its impact on particle picking. Specifically, we selected several publicly available EMPIAR datasets with particle annotations to conduct this evaluation. Fake real micrographs generated by CryoCCD and multiple baselines, were incorporated into the Topaz (Bepler et al., 2019). Subsequently, picked particles were subjected to a standard single-particle reconstruction workflow using CryoSPARC (Punjani et al., 2017) to as-

Table 1: Quantitative comparison of FID and CMMD on six Cryo-EM datasets (lower is better).

| Dataset | TRPV1 | | β-galactosidase | | Rhino/enterovirus | | Innexin-6 | | MLA complex | | GroEL | |
|---|---|---|---|---|---|---|---|---|---|---|---|---|
| | FID↓ | CMMD↓ | FID↓ | CMMD↓ | FID↓ | CMMD↓ | FID↓ | CMMD↓ | FID↓ | CMMD↓ | FID↓ | CMMD↓ |
| Gaussian | 287.04 | 2.81 | 351.27 | 2.96 | 292.56 | 2.74 | 328.17 | 2.38 | 225.20 | 1.80 | 364.28 | 1.87 |
| Poisson | 396.59 | 3.19 | 515.91 | 2.89 | 395.59 | 3.58 | 467.19 | 2.66 | 388.26 | 2.57 | 410.38 | 2.33 |
| Poi–Gau | 290.68 | 2.86 | 356.96 | 3.01 | 301.77 | 2.77 | 272.24 | 2.28 | 239.43 | 1.80 | 369.95 | 1.87 |
| CycleDiffusion | 199.47 | 2.20 | 247.09 | 2.04 | 210.22 | 1.98 | 208.23 | 1.95 | 197.15 | 2.00 | 270.58 | 2.62 |
| CryoGEM | 192.28 | 2.17 | 283.28 | 2.73 | 224.18 | 3.41 | 223.02 | 3.76 | 189.46 | 2.93 | 279.20 | 2.77 |
| CycleNet | 211.30 | 2.33 | 235.02 | 2.09 | 228.79 | 3.10 | 215.66 | 2.47 | 203.50 | 2.18 | 290.41 | 2.86 |
| **Ours** | **121.32** | **1.53** | **137.57** | **1.66** | **130.89** | **1.63** | **130.52** | **1.46** | **88.55** | **0.94** | **147.41** | **1.32** |

Table 2: **Quantitative comparison of particle picking.** Our approach consistently achieves the best in AUPRC↑ and Precision↑ metrics.

| Metric | AUPRC↑ | | | | Precision↑ | | | |
|---|---|---|---|---|---|---|---|---|
| Dataset | Proteasome | Integrin | PhageMS2 | HumanBAF | Proteasome | Integrin | PhageMS2 | HumanBAF |
| Gaussian | 0.463 | 0.233 | 0.575 | 0.449 | 0.412 | 0.204 | 0.538 | 0.397 |
| Poisson | 0.458 | 0.195 | 0.408 | 0.392 | 0.386 | 0.168 | 0.365 | 0.341 |
| Poi–Gau | 0.438 | 0.244 | 0.601 | 0.376 | 0.397 | 0.227 | 0.519 | 0.347 |
| CycleDiffusion | 0.357 | 0.233 | 0.420 | 0.369 | 0.321 | 0.196 | 0.347 | 0.326 |
| CryoGEM | 0.257 | 0.208 | 0.159 | 0.178 | 0.221 | 0.175 | 0.148 | 0.159 |
| CycleNet | 0.343 | 0.251 | 0.399 | 0.297 | 0.307 | 0.234 | 0.340 | 0.303 |
| Topaz | 0.301 | 0.510 | 0.317 | 0.487 | 0.279 | 0.452 | 0.286 | 0.462 |
| **Ours** | **0.479** | **0.532** | **0.797** | **0.588** | **0.437** | **0.497** | **0.736** | **0.618** |

sess the resulting structural resolution improvements. Qualitatively, particle-picking visualizations demonstrated that Topaz models finetuned with CryoCCD-generated noisy synthetic data provided superior discrimination between true particles and background noise. Improvements were confirmed through the Area Under the Precision-Recall Curve (AUPRC) metric (Bepler et al., 2019; Wagner et al., 2019), where our method consistently outperformed all other baselines. Additionally, corresponding enhancements in final reconstruction resolutions (measured in Ångströms) further validated the practical utility of CryoCCD. Figure 8 in Appendix C.1 clearly shows our visualization results. Analysis and metrics are also comprehensively documented in Table 2 and Appendix C.1.

## 5.3 POSE ESTIMATION

Accurate particle orientation determination represents a critical bottleneck in achieving high-resolution cryo-EM reconstructions. We evaluated CryoCCD's capacity to enhance pose estimation by applying our noisy synthetic datasets to train state-of-the-art ab-initio reconstruction methods CryoFIRE (Levy et al., 2022). The synthetic particles, complete with precise orientation annotations, provided an ideal training corpus for this supervised learning paradigm. We subsequently assessed performance on real micrographs using Filter Back-Projection (FBP) as our reconstruction algorithm. Our method demonstrated improvements over other baselines, achieving higher rotation accuracy and enhanced resolution quality across several datasets. Table 3 presents the pose estimation results, while Figure 9 in Appendix C.2 provides the corresponding FSC curves.

Table 3: **Quantitative comparison of pose estimation.** Our approach achieves the best in Res(px)↓ and Rot.(rad)↓ metrics.

| Metric | Res(px)↓ | | | | Rot.(rad)↓ | | | |
|---|---|---|---|---|---|---|---|---|
| Dataset | Proteasome | Integrin | PhageMS2 | HumanBAF | Proteasome | Integrin | PhageMS2 | HumanBAF |
| Gaussian | 3.03 | 7.01 | 5.96 | 7.08 | 0.51 | 1.49 | 0.99 | 1.58 |
| Poisson | 3.12 | 7.14 | 6.09 | 9.33 | 1.09 | 1.66 | 1.03 | 1.60 |
| Poi–Gau | 3.15 | 7.99 | 5.98 | 9.09 | 1.32 | 1.57 | 0.73 | 1.51 |
| CycleDiffusion | 3.67 | 9.04 | 6.33 | 9.02 | 0.47 | 1.41 | 0.97 | 1.67 |
| CryoGEM | 7.99 | 9.24 | 15.38 | 10.79 | 1.72 | 1.94 | 1.20 | 1.78 |
| CycleNet | 4.58 | 8.31 | 9.04 | 8.91 | 1.03 | 1.50 | 1.19 | 1.69 |
| CryoFIRE | 6.02 | 13.27 | 18.03 | 7.17 | 1.54 | 0.97 | 0.75 | 1.49 |
| **Ours** | **2.87** | **5.89** | **5.37** | **7.01** | **0.44** | **0.90** | **0.51** | **1.47** |

## 5.4 GENERALIZATION TO HELD-OUT PROTEIN FAMILIES

To demonstrate the generalization ability of CryoCCD, We evaluate CryoCCD on four EMPIAR datasets that were not used during training— EMPIAR-10028 (80S ribosome) (Wong et al., 2014), EMPIAR-10059 (TRPV1) (Gao et al., 2016), EMPIAR-10389 (urease) (Righetto et al., 2020), and EMPIAR-10532 (FANCD2–FANCI) (Tan and Rubinstein, 2020). For each dataset we run the full pipeline and report FID and CMMD. As shown in Table 4, CryoCCD consistently achieves the lowest scores against Gaussian, Poisson, Gaussian–Poisson mixture, CryoGEM, and CycleNet, indicating strong realism and generalization to unseen protein families.

Table 4: Generalization to held-out protein families: FID↓ and CMMD↓ on four EMPIAR datasets not used during training.

| Dataset | EMPIAR-10028 | | EMPIAR-10059 | | EMPIAR-10389 | | EMPIAR-10532 | |
| --- | --- | --- | --- | --- | --- | --- | --- | --- |
| | FID↓ | CMMD↓ | FID↓ | CMMD↓ | FID↓ | CMMD↓ | FID↓ | CMMD↓ |
| Gaussian | 290.34 | 2.85 | 301.07 | 2.90 | 275.46 | 2.62 | 299.01 | 2.87 |
| Poisson | 443.13 | 3.43 | 389.78 | 3.31 | 391.27 | 3.36 | 410.98 | 3.50 |
| Poi–Gau | 286.46 | 2.77 | 300.98 | 2.87 | 277.04 | 2.64 | 307.87 | 2.91 |
| CryoGEM | 232.69 | 2.01 | 247.58 | 2.08 | 244.33 | 2.05 | 280.61 | 2.74 |
| CycleNet | 210.47 | 1.94 | 227.48 | 2.03 | 230.09 | 1.81 | 222.56 | 2.10 |
| **Ours** | **135.77** | **1.67** | **140.54** | **1.79** | **131.90** | **1.60** | **158.75** | **1.84** |

Table 5: Quantitative evaluation of the ablation study on diffusion components and biophysical modeling components, reported in FID↓ across six datasets.

| Setting | TRPV1 | $\beta$-gal. | Rhino | Innexin-6 | MLA | GroEL |
| --- | --- | --- | --- | --- | --- | --- |
| w/o cycle consistency | 243.38 | 254.97 | 270.01 | 248.53 | 199.79 | 300.35 |
| w/o mask-guided contrastive learning | 155.95 | 163.31 | 159.44 | 169.01 | 122.57 | 191.49 |
| Ice layer: Global-gradient approach | 199.84 | 270.32 | 212.09 | 228.65 | 197.52 | 284.43 |
| CTF: small defocus | 149.33 | 151.79 | 163.31 | 147.28 | 104.01 | 180.43 |
| CTF: large defocus | 148.47 | 156.90 | 166.44 | 150.80 | 107.09 | 184.47 |
| **Ours** | **121.32** | **137.57** | **130.89** | **130.52** | **88.55** | **147.41** |

## 5.5 ABLATION STUDY

**Diffusion model components.** We further ablate two learning modules of the reconstruction model: *cycle consistency* and *mask-guided contrastive learning*. Table 5 reports FID on six datasets. Removing cycle consistency substantially worsens FID and disrupts particle localization after noise augmentation; removing mask-guided contrastive learning also degrades FID and yields blurred boundaries with poor foreground–background separation. We include visual comparisons in the supplement to illustrate these effects.

**Biophysical modeling.** To assess the impact of our physics-informed imaging components, we performed a quantitative ablation study focusing on the *ice layer* model and the *CTF* parameterization. We found that both elements are crucial for high-fidelity image simulation. We show the results in Table 5 and detail the analysis in Appendix D.1.

**Samplers and Sampling Steps.** We further analyze the effect of different diffusion sampling steps and samplers. Increasing the step budget from 5 to 50 yields a clear FID improvement. The complete quantitative results are reported in Appendix D.2.

## 6 CONCLUSION

In this paper, we introduce CryoCCD, a synthesis framework that integrates biophysical modeling with conditional cycle-consistent diffusion to generate structurally diverse and noise-realistic cryo-EM micrographs. Our method achieves state-of-the-art performance on FID, CMMD, and two downstream tasks across six public datasets. Moreover, CryoCCD exhibits strong generalization to held-out protein families. We hope this thorough and high-fidelity simulator will substantially reduce the annotation burden for biologists and accelerate the development of computational tools for the cryo-EM community.

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

# A DETAILS OF BIOPHYSICAL MODELING

## A.1 IMPLEMENTATION DETAILS

**Structure Library Construction.** The structure-based mining targets cryo-EM-derived entries from the Protein Data Bank using a resolution cutoff <4.0 Å with molecular weight and symmetry classification spanning biological assemblies across diverse size scales and point group symmetries. AlphaFold3 integration addresses structural coverage gaps and increases heterogeneity by generating predicted models for sequences that lack experimental structures, while expanding conformational heterogeneity by predicting and modeling structures from sequences with potential conformational diversity. The structures are processed through confidence-stratified sampling: very high confidence regions (pLDDT >90) remain static, confident regions (70-90) undergo constrained perturbations, low confidence regions (50-70) experience enhanced sampling, and very low confidence regions (<50) are treated as fully flexible. Interdomain flexibility employs PAE-guided rigid-body sampling for regions with PAE >10 Å, generating structural ensembles that capture prediction uncertainty while preserving fold topology. The pipeline for enhancing compositional and conformational heterogeneity is illustrated in Figure 5.

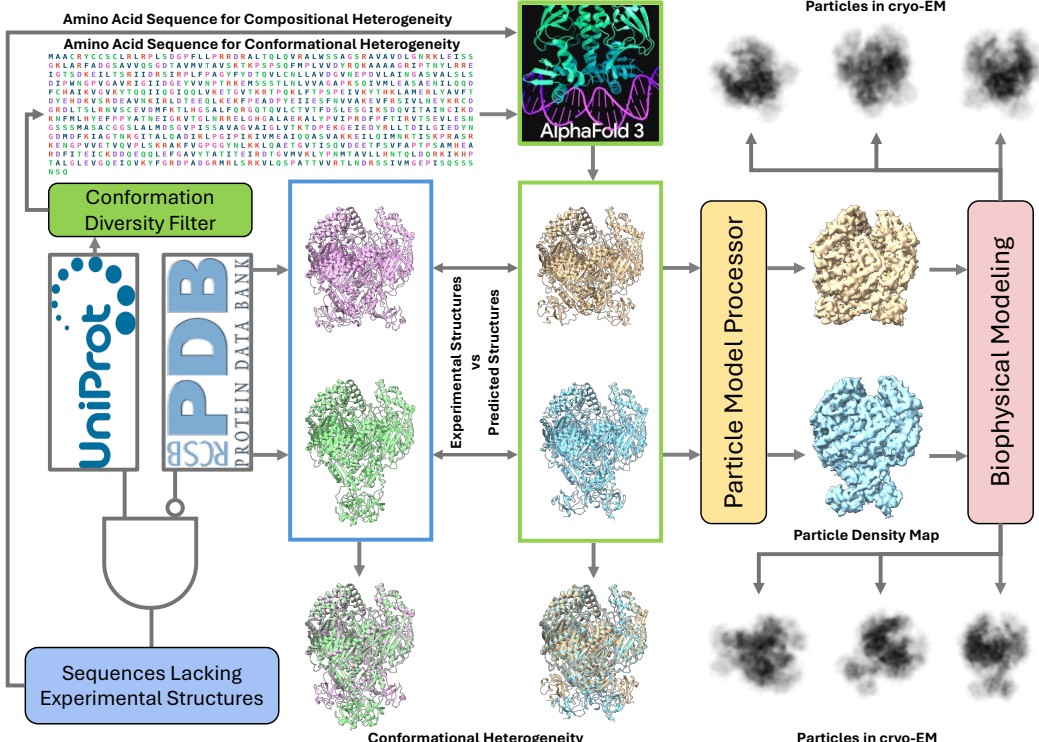

Figure 5: **AlphaFold3-based pipeline for enhancing compositional and conformational heterogeneity of the structure library.** Sequences lacking experimental structures are selected from the UniProt (Consortium, 2019) database to increase compositional heterogeneity, and entries with potential conformational diversity are selected to enhance conformational heterogeneity. The green frame shows the atomic models of two human PNPase (Q8TCS8) conformations processed based on AlphaFold3; in the blue frame, for comparison, are the open formation (9KJR) and closed formation (9KJT) of human PNPase from the Protein Data Bank. Below the frames are illustrations of the differences between these conformations. The generated particles are then converted into density maps and undergo biophysical modeling to generate simulated cryo-EM images.

**Atomic Model Conversion.** The atomic model to volume conversion employs element-specific van der Waals radii based on established crystallographic databases, with different radii assigned to reflect the electronic structure and coordination environments of various atomic species. For

unrecognized elements, a conservative default radius is applied based on typical organic atom dimensions. The Gaussian density kernel utilizes width $\sigma = R_{vdW}/(2 \cdot \text{spacing})$ where voxel spacing follows spacing $= \text{resolution}/2.0$ to ensure Nyquist-compliant sampling. Box margin calculation applies margin $= \max(3R_{max}, 2 \cdot \text{resolution})$ with automatic expansion by $4R_{max}$ when boundary violations exceed 10% of total atoms. Post-processing includes Gaussian smoothing with $\sigma_{smooth} = \text{resolution}/(2.0 \cdot \text{spacing})$ and background thresholding at 0.005 relative intensity to eliminate noise while preserving structural features. Examples of the resulting structure density volumes are illustrated in Figure 6.

**Scale-Adaptive Parameterization.** The automatic scale detection employs size-based classification with specific thresholds, particles <10 Å receive scale factor $s = 1.0$, 10-50 Å receive $s = 0.8$, 50-200 Å receive $s = 0.6$, 200-1000 Å receive $s = 0.4$, and >1000 Å receive $s = 0.2$. The composite scale factor follows $s = 0.7s_{size} + 0.3s_{density}$ where density component reflects local crowding through $s_{density} = \min(1.0, \text{particle\_size}^3/(\text{volume\_size}/1000))$. Scale-dependent parameter relationships include overlap threshold $= 0.4 - 0.3s$, placement density $= 0.7 + 0.5s$, collision strictness $= 0.5 + 0.5s$, and mesh reduction factor $= 0.7 - 0.7s$. These scaling laws ensure appropriate geometric constraints while optimizing computational resources across biological size ranges. Figure 6 illustrates the generated 3D volumes rendered at varying scales.

**Distribution Modeling.** The placement implementation combines data-driven coordinates from RELION with synthetic insertion algorithms. Data-driven placement converts 2D picks $(x, y)$ to 3D translations $T_{\text{exp}}$ through coordinate system transformation, while Euler angles $(\alpha, \beta, \gamma)$ undergo conversion to quaternions $q_{\text{exp}}$ using standard attitude conversion conventions. Synthetic insertion employs multiple distribution strategies, uniform placement uses rejection sampling with collision avoidance radius $2R_{particle}(1 - \text{overlap\_threshold})$ and maximum iteration limits; Gaussian clustering utilizes standard deviations $\sigma_{primary} = \min(\text{volume\_shape})/6$ for concentrated regions and $\sigma_{secondary} = 0.7\sigma_{primary}$ for peripheral zones; grid placement applies spacing $d = 2R_{particle}(1 - \text{overlap\_threshold}/2)$ with positional jitter $\pm d/4$; interface placement constrains particles within distance tolerance from target surfaces. Class-specific rules implement distinct organizational patterns. For example, soluble enzymes follow uniform dispersion, ribosomes cluster according to measured inter-particle distances to mimic polysomes, and viral capsids follow separation distributions to avoid unrealistic aggregation.

**Orientational Sampling.** Particle orientation sampling employs multiple strategies reflecting different biophysical constraints. Uniform sampling draws from rotation group $SO(3)$ using quaternion parameterization for isotropic coverage. Preferred axis alignment utilizes von Mises-Fisher distributions $f(\mathbf{x}; \boldsymbol{\mu}, \kappa) = \kappa \exp(\kappa \boldsymbol{\mu}^T \mathbf{x})/(4\pi \sinh(\kappa))$ with concentration parameter $\kappa = 10$ providing moderate orientational bias suitable for air-water interface effects. Limited tilt sampling constrains orientations within $\theta_{max} = \pi/6$ from vertical using truncated normal distributions, reflecting adsorption-induced preferences observed in cryo-EM. Numerical stability ensures quaternion normalization within appropriate tolerance with iterative renormalization when necessary.

**Mesh Generation and Geometric Processing.** Surface mesh construction employs marching cubes algorithm with adaptive step sizing, unit steps for high-resolution simulations ($s > 0.8$) and increased step sizes for coarse applications. Triangle mesh quality maintenance includes aspect ratio constraints, minimum angle thresholds, and edge length limits to ensure geometric fidelity. Smoothing operations apply $N_{iterations} = \max(1, 5s)$ iterations with relaxation factor 0.2. Mesh decimation utilizes topology-preserving edge collapse when reduction factor $0.6(1 - s)$ exceeds threshold values. Surface normal computation employs area-weighted vertex averaging to maintain geometric accuracy. Collision detection uses octree structures with adaptive depth and leaf capacity based on scale factor, enabling efficient proximity queries during placement operations.

**Ice-Layer Modeling.** Vitrified ice simulation incorporates experimentally-derived parameters for realistic specimen conditions. Thickness distribution follows log-normal parameterization with $\mu = \ln(100)$ and $\sigma = 0.2$, yielding mean thickness 100 nm with variance matching experimental observations. Surface topography employs multi-octave Perlin noise with amplitude scaling $A_i = 5.0/2^i$ nm and fundamental wavelength $L = 10$ pixels, capturing fractal roughness characteristics observed in vitrified specimens. Density fluctuations apply spatially correlated Gaussian

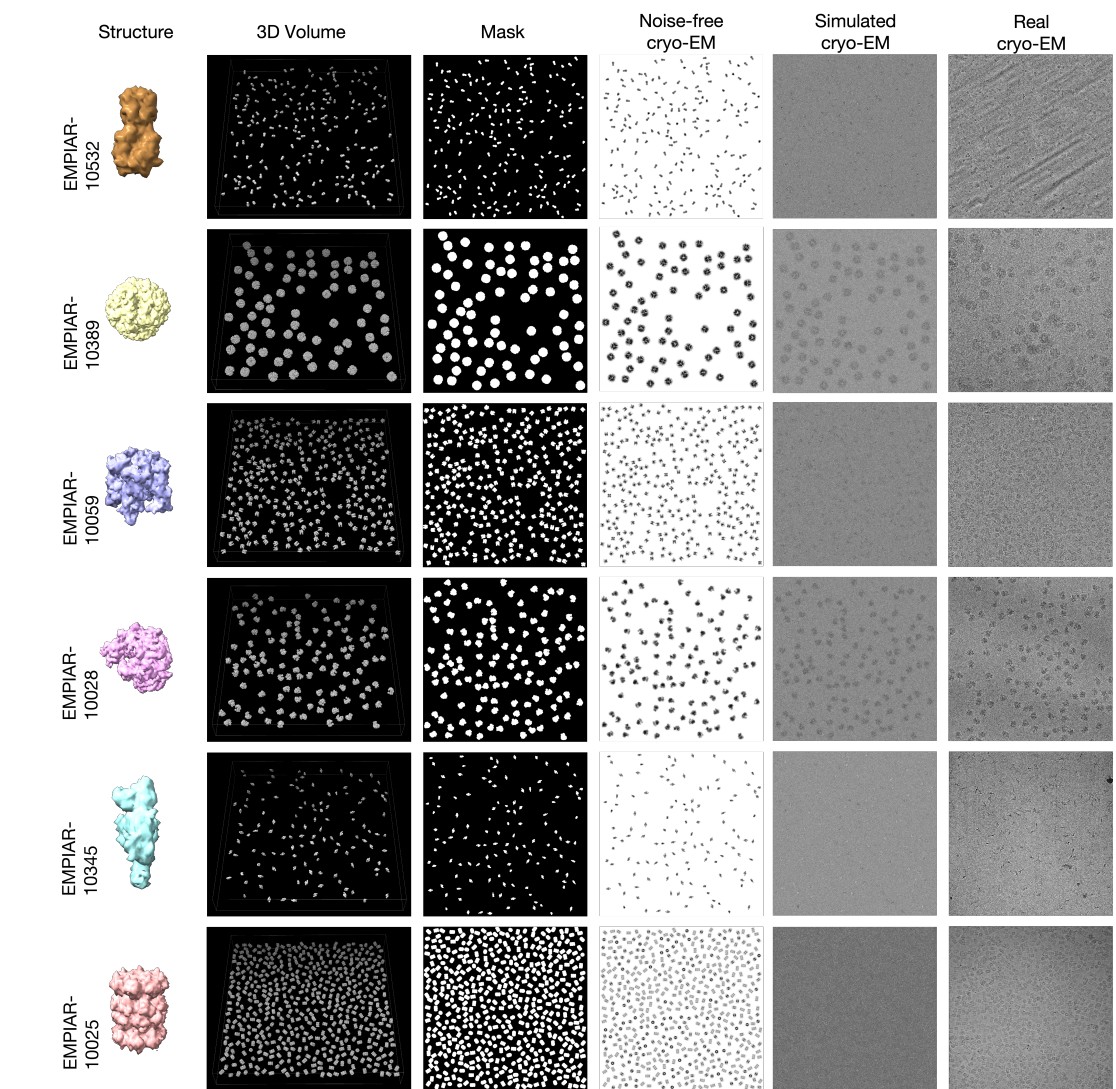

Figure 6: Visual Examples from the Cryo-EM Biophysical Simulation Pipeline.

noise with amplitude $\sigma = 0.05\rho_{ice}$ and correlation length appropriate for amorphous ice structure. The ice density $\rho_{ice} = 0.92$ g/cm³ represents vitrified water at cryogenic temperatures, with surface height constraints maintaining reasonable thickness ranges to prevent unrealistic topographic extremes.

**Electrostatic Potential Assembly and Projection.** The electrostatic potential calculation follows $\Phi(\mathbf{r}) = \sum_{i=1}^{L} \rho_i(R(q_i)^{-1}(\mathbf{r} - T_i))$ with atomic scattering factors incorporating appropriate approximations and relativistic corrections for high-voltage applications. Weak-phase projection $P(x, y) = \int_{-z_0/2}^{z_0/2} \Phi(x, y, z) \, dz$ employs numerical integration with adaptive step sizing. The contrast transfer function uses defocus values optimized for different resolution targets; in the Fourier domain, the CTF is $H(s, a) = -\left(\sqrt{1 - w^2} \sin \gamma(s, a) - w \cos \gamma(s, a)\right) \exp\left(-\frac{Bs^2}{4}\right)$ with $\gamma(s, a) = \frac{\pi}{2}(2\lambda \Delta Z \, s^2 + \lambda^3 C_s \, s^4) - \phi$, $s = \sqrt{k_x^2 + k_y^2}$, and $a = \text{atan2}(k_y, k_x)$. Parameter specifications include spherical aberration $C_s = 2.7$ mm for conventional systems, amplitude contrast $w = 0.07$ for biological specimens, and B-factors reflecting radiation sensitivity and molecular flexibility. The phase function incorporates electron wavelength $\lambda = 12.2643247/\sqrt{kV(1 + 0.978466 \times 10^{-6} \cdot kV)}$ with relativistic corrections for high-voltage appli-

cations. An inverse Fourier transform of the filtered spectrum yields a noise-free micrograph used as the clean input for diffusion-based noise synthesis.

## A.2 VISUALIZATION OF THE CRYO-EM SIMULATION PIPELINE

As shown in Figure 6, we visualized the pipeline for biophysically simulating synthetic cryo-EM data. The leftmost column shows real cryo-EM micrographs from the EMPIAR database. The corresponding extracted structures are used as inputs, and the adjacent column to the right displays their processed density volumes. These structures are then inserted into 3D virtual sample volumes to mimic realistic molecular distributions, and binary masks are computed to indicate spatial occupancy. The final column displays noise-free simulated cryo-EM images generated through biophysical modeling, which serve as structurally diverse clean inputs for subsequent CryoCCD training or inference.

# B EXPERIMENT SETTINGS

## B.1 DATASET DETAILS

This section provides detailed information about EMPIAR datasets utilized in our experiments. These datasets represent a diverse range of molecular structures with various characteristics that present unique challenges for cryo-EM reconstruction. Among these datasets, 15 datasets serve as training data (EMPIAR-10005, 10061, 10083, 10199, 10289, 10363, 10366, 10421, 10425, 10431, 10551, 12003, 12330, 12598, 12667), which we merge into a single, large-scale corpus so that the model is exposed to diverse imaging conditions and can learn features that generalize beyond any one specimen. Six additional datasets are reserved exclusively for visual-quality evaluation: TRPV1 (EMPIAR-10005), $\beta$-galactosidase (EMPIAR-10061), Rhino/enterovirus (EMPIAR-10199), Innexin-6 (EMPIAR-10289), MLA complex (EMPIAR-10425), and GroEL (EMPIAR-12667). Additionally, four publicly available cryo-EM datasets (EMPIAR-10025, 10075, 10345, and 10590) are used to validate our framework on the downstream tasks of particle picking and pose estimation. EMPIAR-10028 (80S ribosome), EMPIAR-10059 (TRPV1), EMPIAR-10389 (urease), and EMPIAR-10532 (FANCD2–FANCI) are used as held-out protein datasets to demonstrate the generalization ability of CryoCCD.

**EMPIAR-10005.** Rat TRPV1 ion channel (Liao et al., 2013) (EMPIAR-10005) achieved a high-resolution result at 3.4 Å, breaking the side-chain resolution barrier for membrane proteins without crystallization. The structure exhibits four-fold symmetry around a central ion pathway formed by transmembrane segments 5-6 and the intervening pore loop, with a wide extracellular mouth and a short selectivity filter.

**EMPIAR-10025.** The Thermoplasma acidophilum 20S proteasome (Campbell et al., 2015) (EMPIAR-10025) dataset comprises 196 real micrographs and 49,954 manually curated particles, enabling reconstruction at a high resolution of 2.8 Å. The T20S proteasome is a 700 kDa complex composed of 14 $\alpha$-subunits and 14 $\beta$-subunits arranged with D7 symmetry. The micrographs contain particles exhibiting high symmetry, random in-plane orientations, and frequent mutual occlusion, posing challenges for particle picking.

**EMPIAR-10028.** *Plasmodium falciparum* 80S ribosome bound to the anti-protozoan drug emetine (Wong et al., 2014) (EMPIAR-10028) was determined by cryo-EM at a resolution of 3.2 Å. The structure revealed drug–ribosome interactions in malaria parasites, with related PDB entries 3j79 and 3j7a and EMDB entry EMD-2660. The dataset contains 1081 multi-frame micrographs (4096×4096 pixels, 16 frames each) and 105,247 processed particles, totaling 1.2 TB. These data enabled detailed structural analysis of the malaria ribosome and its inhibition by small molecules.

**EMPIAR-10059.** TRPV1 ion channel in complex with DkTx and RTX, embedded in lipid nanodiscs (Gao et al., 2016) (EMPIAR-10059) was determined by cryo-EM at a resolution of 2.95 Å. The structure (PDB 5irx; EMDB EMD-8117) revealed mechanisms of ligand and lipid modulation of TRPV1. The dataset contains 1200 motion-corrected micrographs (3838×3710 pixels, 30-frame exposures) and 218,805 raw particle images (192×192 pixels), totaling 93.8 GB.

**EMPIAR-10061.** *E. coli* $\beta$-galactosidase with PETG inhibitor (Bartesaghi et al., 2015) was determined by cryo-EM at an average resolution of 2.2 Å. The map contains identifiable densities for approximately 800 water molecules and for magnesium and sodium ions, demonstrating that specimen preparation quality and protein flexibility are now the major bottlenecks to routinely achieving near 2 Å resolutions.

**EMPIAR-10075.** The MS2 bacteriophage(Koning et al., 2016) (EMPIAR-10075) dataset contains 300 real micrographs and 12,682 manually curated particles, enabling a reconstruction at 8.7 Å resolution. The virus comprises 178 copies of the coat protein, a single A-protein, and an encapsulated RNA genome. Due to its large size and structural variability, PhageMS2 presents a significant challenge for particle detection and classification.

**EMPIAR-10083.** P22 bacteriophage capsid (Hryc et al., 2017) presents a 3.3-Å map of this large and complex macromolecular assembly. The dataset demonstrates a computational procedure to derive an atomic model with annotated metadata, identifying previously undescribed molecular interactions between capsid subunits that maintain stability without cementing proteins or cross-linking found in other bacteriophages.

**EMPIAR-10199.** Rhino- and enterovirus capsid (Abdelnabi et al., 2019) revealed a previously unknown druggable pocket formed by viral proteins VP1 and VP3, conserved across entero-/rhinovirus species. This discovery led to the identification of inhibitor analogues with broad-spectrum activity against multiple virus groups, providing novel insights into viral entry biology.

**EMPIAR-10289.** *C. elegans* innexin-6 gap junction proteins (Burendei et al., 2020) shows the structures in an undocked hemichannel form. In the nanodisc-reconstituted structure of the wild-type INX-6 hemichannel, flat double-layer densities obstruct the channel pore, revealing insights into lipid-mediated amino-terminal rearrangement and pore obstruction upon nanodisc reconstitution.

**EMPIAR-10345.** The asymmetric $\alpha V\beta 8$ integrin (Cormier et al., 2018) (EMPIAR-10345) dataset consists of 1,644 micrographs and 84,266 manually selected particles, yielding a 3.3 Å resolution structure. This complex features the human $\alpha V\beta 8$ ectodomain bound to porcine L-TGF-$\beta 1$, and is characterized by significant conformational flexibility, particularly in the leg domain, as observed in the micrographs.

**EMPIAR-10363.** Single-particle reconstruction with aberration correction (Bromberg et al., 2020) provides an analysis of how uncorrected antisymmetric aberrations affect cryo-EM results. The reference-based aberration refinement for two datasets acquired with a 200 kV microscope in the presence of significant coma yielded 2.3 and 2.7 Å reconstructions for 144 and 173 kDa particles, respectively.

**EMPIAR-10366.** Canine distemper virus F protein (Kalbermatter et al., 2020) presents the prefusion state of the CDV F protein ectodomain at 4.3 Å resolution. Stabilization was achieved by fusing the GCNt trimerization sequence at the C-terminal region and purifying with a morbilliviral fusion inhibitor, with the 3D map showing clear density for the ligand at the protein binding site.

**EMPIAR-10389.** Urease from the pathogen *Yersinia enterocolitica* (Righetto et al., 2020) (EMPIAR-10389) was determined by cryo-EM at 1.98 Å resolution (PDB 6yl3; EMDB EMD-10835). The high-resolution structure revealed detailed features of the multimeric urease complex and provided insights into its pathogenic function. The dataset contains motion-corrected micrographs, totaling 839.9 GB.

**EMPIAR-10421.** High-speed specimen preparation technique (Tan and Rubinstein, 2020) demonstrates a method where solution sprayed onto one side of a holey cryo-EM grid is wicked through by a glass-fiber filter held against the opposite side. This "Back-it-up" (BIU) approach produces suitable films for vitrification in tens of milliseconds, creating large areas of ice suitable for both soluble and detergent-solubilized protein complexes.

**EMPIAR-10425.** *A. baumannii* MLA complex (Mann et al., 2021) presents cryo-EM maps of the core MlaBDEF complex in apo-, ATP- and ADP-bound states. The maps reveal multiple lipid binding sites in both cytosolic and periplasmic sides of the complex, with molecular dynamics simulations suggesting potential lipid trajectories across the membrane.

**EMPIAR-10431.** Human CDK-activating kinase (CAK) (Greber et al., 2020) contains the three-dimensional structure of this critical regulator of transcription initiation and the cell cycle. The dataset includes both the catalytic module structure and CAK in complex with covalently bound inhibitor THZ1, providing insights into assembly, CDK7 activation, and inhibitor binding for therapeutic compound design.

**EMPIAR-10532.** Influenza hemagglutinin (HA) trimer vitrified using the Back-it-up (BIU) device (Tan and Rubinstein, 2020) (EMPIAR-10532) was determined by cryo-EM at 2.9 Å resolution (PDB 6wxb; EMDB EMD-21954). The study demonstrated that through-grid wicking enables high-speed cryo-EM specimen preparation. The dataset contains 1,556 raw Falcon IV movies (4096×4096 pixels, 30 frames each), the corresponding aligned micrographs, and 128,305 refined particle images (256×256 pixels) with assigned Euler angles and shifts, totaling 1.2 TB.

**EMPIAR-10551.** Adeno-associated virus AAV-DJ (Xie et al., 2020) was determined to 1.56 Å resolution, nearly matching the highest resolutions ever attained through X-ray diffraction of virus crystals. At this exceptional resolution, most hydrogens are clearly visible, improving atomic refinement accuracy and revealing that hydrogen bond networks are quite different from those inferred at lower resolutions.

**EMPIAR-10590.** The endogenous human BAF complex(Mashtalir et al., 2020) (EMPIAR-10590) includes 300 micrographs and 62,493 manually picked particles, achieving a reconstruction resolution of 7.8 Å. This complex is known for its compositional and conformational heterogeneity, which increases the difficulty of automated particle picking.

**EMPIAR-12003.** Human pseudouridine synthase 3 (Lin et al., 2024) presents structures in apo form and bound to three tRNAs, showing how the symmetric PUS3 homodimer recognizes tRNAs and positions the target uridine. Combined with structure-guided mutations and transcriptome-wide $\Psi$ site mapping, this dataset provides the molecular basis for PUS3-mediated tRNA modification and its link to intellectual disabilities.

**EMPIAR-12330.** Yeast spliceosome with Fyv6 (Senn et al., 2024) contains a high-resolution (2.3 Å) structure of a product complex spliceosome. The structure reveals Fyv6 as the only second step factor contacting the Prp22 ATPase, with Fyv6 binding mutually exclusive with first step factor Yju2, supporting a model where their exchange facilitates exon ligation.

**EMPIAR-12598.** SARS-CoV-2 RdRp backtracking (Malone et al., 2021) uses cryo-EM, RNA-protein cross-linking, and molecular dynamics simulations to characterize viral RNA polymerase behavior. The results establish the mechanism by which the product RNA extrudes through the RdRp nucleoside triphosphate entry tunnel during backtracking, a process that may aid in proofreading and contribute to viral resistance against antivirals.

**EMPIAR-12667.** GroEL chaperonin with inhibitors (Godek et al., 2024) employed cryo-EM to establish the binding site of bis-sulfonamido-2-phenylbenzoxazoles at the GroEL ring-ring interface. The biochemical characterization showed potent inhibition of Gram-negative chaperonins but lower potency against Gram-positive organisms, validating this chaperone system as an antibiotic target against bacteria including ESKAPE pathogens.

### B.2 Baseline Details

**Traditional Noise Simulations.** These baselines include Gaussian, Poisson, and Poisson-Gaussian (Poi–Gau) noise simulations, representing commonly used traditional strategies in cryo-EM synthetic data generation. Following Vulovic et al. (Vulović et al., 2013), the Gaussian noise baseline introduces zero-mean additive Gaussian noise to simulate a mixture of readout noise, dark current, and background structural variations. The Poisson noise baseline captures the stochastic nature of electron detection under low-dose imaging conditions, reflecting quantum noise characteristics. The Poi–Gau baseline combines both noise types to model complex imaging artifacts. For fair comparison, the Signal-to-Noise Ratio (SNR) is uniformly set to 0.1 across all three methods.

**CycleDiffusion (Wu and De la Torre, 2022).** This method employs two diffusion models trained independently on synthetic and real cryo-EM domains, respectively. Each DDIM is trained on 500 images of resolution $512 \times 512$ over 10,000 steps with a batch size of 8. The translation from

synthetic to real is achieved by running a cycle-consistent sampling loop without paired supervision. We use the publicly available implementation and default inference settings.

**CryoGEM (Zhang et al., 2024).** CryoGEM is a physics-informed generative model designed to produce synthetic cryo-EM images with realistic noise characteristics. To introduce authentic noise, CryoGEM employs an unpaired noise translation technique that leverages contrastive learning with a novel mask-guided sampling strategy. This approach effectively transforms the simulated images into ones with realistic noise patterns. All experiments are performed with the method's default settings.

**CycleNet (Xu et al., 2023).** CycleNet enables unpaired image-to-image translation with pre-trained diffusion models by enforcing cycle-consistency regularization. Built upon ControlNet (Zhang et al., 2023) and Stable Diffusion (Rombach et al., 2022), it performs a forward translation from source to target domain and a backward reconstruction to ensure semantic preservation. Compared to mask- or attention-based approaches, CycleNet achieves higher consistency and translation faithfulness, and demonstrates strong zero-shot generalization with limited training data.

### B.3 IMPLEMENTATION DETAILS

All experiments are conducted on a single NVIDIA GeForce RTX A100 GPU with 80 GB of memory. The models are trained for 50 epochs using the Adam optimizer with a learning rate of $1 \times 10^{-4}$ and a batch size of 4. To explore the effects of different generative sampling strategies, we evaluate four diffusion samplers within the CryoCCD framework: DDIM (Song et al., 2022), DDPM (Ho et al., 2020), and their accelerated variants, DPM-Solver (Lu et al., 2022), DPM-Solver++ (Lu et al., 2025), UniPC (Zhao et al., 2023), and linear multistep (LMS) sampler (Liu et al., 2022). These sampling methods are examined to understand the trade-off between synthesis quality and inference speed under various configurations.

Our synthetic data generation process begins with biophysical modeling to simulate noise-free cryo-EM micrographs. Given a real cryo-EM image, the corresponding simulated noise-free counterpart and a structural mask, the CryoCCD framework performs inference to generate realistic noisy images that closely mimic experimental conditions. This design enables controllable and interpretable simulation of complex imaging artifacts.

The training dataset is detailed in Section B.1, and we validate the utility of our generated data in two downstream tasks—particle picking (Section C.1) and pose estimation (Section C.2)—demonstrating the effectiveness and generalizability of the proposed framework.

### B.4 EFFICIENCY AND COMPUTATIONAL COMPLEXITY

We benchmark training time, inference latency, and peak memory on a single NVIDIA A100 GPU to clarify computational feasibility. As expected for a lightweight GAN, CryoGEM attains the fastest speed and lowest memory, but suffers from instability and eventual collapse in our mixed datasets (see the CryoGEM column in Fig. 4). Compared with Cy-

Figure 7: Efficiency on a single NVIDIA A100: training time (s/epoch), inference time (s/img), peak memory (MB).

| Method | Train | Infer | Memory |
|--------|-------|-------|--------|
| CryoGEM | 209 | 0.2 | 4150 |
| CycleDiffusion | 1267 | 3.1 | 54377 |
| Ours | 805 | 1.9 | 68341 |

cleDiffusion, CryoCCD reduces training time (805 s vs. 1267 s per epoch) and inference latency (1.9 s vs. 3.1 s per image) while yielding better visual and downstream performance. The efficiency gains stem from mask-guided contrastive learning and a lightweight discriminator, which provide strong structural guidance so fewer diffusion steps suffice. Our peak memory is higher due to these additional modules.

## C  DETAILS OF DOWNSTREAM TASK

### C.1  PARTICLE PICKING

Topaz (Bepler et al., 2019) is a cryo-EM particle picking framework that assigns each pixel a probability of belonging to a particle. A user-defined threshold is then applied to retain high-confidence detections. We adopt a pre-trained Topaz model as our baseline without further fine-tuning.

**Evaluation via AUPRC.**  To compare different picking methods, we rank detection candidates by their confidence scores and select the top $N$ predictions for analysis. Varying a threshold set $\{\tau_i\}_{i=1}^{n-1}$ partitions the confidence range into $n$ levels, allowing computation of precision $\Pr(k)$ and recall $\Re(k)$ at each level. The area under the resulting precision–recall curve (AUPRC) is defined as

$$\text{AUPRC} = \sum_{k=1}^{n} \Pr(k)\big(\Re(k) - \Re(k-1)\big),\tag{13}$$

where the precision $\Pr(k)$ is the fraction of true positives among predictions with probability $\geq \tau_k$, and the recall $\Re(k)$ is the fraction of true positives with probability $\geq \tau_k$ relative to the total number of true positives.

**Evaluation via Precision.**  Beyond AUPRC, we also report the overall precision at a fixed confidence cutoff. Precision is computed as

$$\text{Precision} = \frac{\text{TP}}{\text{TP} + \text{FP}}\tag{14}$$

where TP and FP denote true and false positives, respectively, at the chosen threshold. This metric directly reflects the accuracy of detected particles without regard to recall.

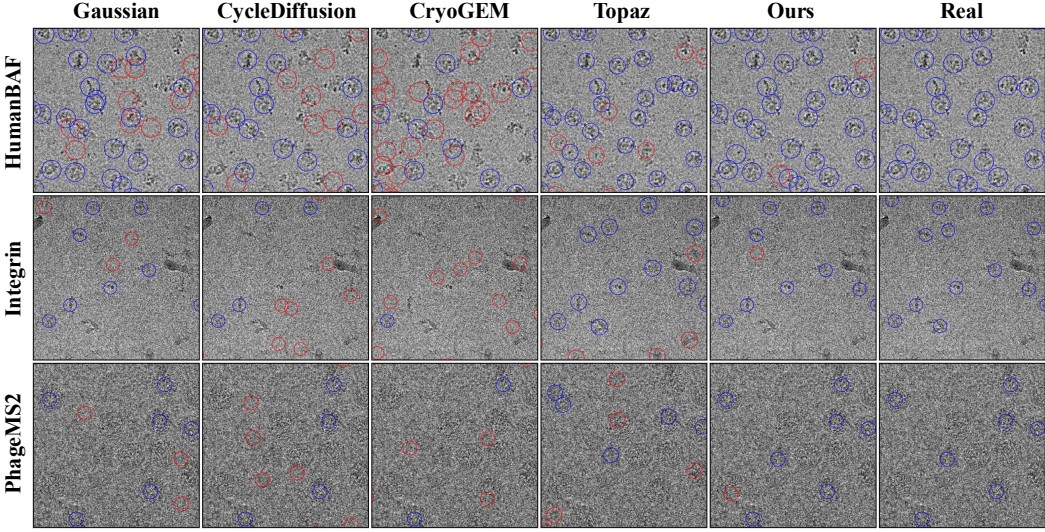

Figure 8: **Qualitative comparison results of particle picking.** The blue circles indicate matches with manual picking results, while the red circles represent misses or excess picks by the model.

**Result Analysis.**  Table 2 summarizes both AUPRC and overall precision for each method. Overall, our detector exhibits consistent improvements over classical noise-model baselines (Gaussian, Poisson, Hybrid) as well as recent learning-based approaches (CycleDiffusion (Wu and De la Torre, 2022), CryoGEM (Zhang et al., 2024), Topaz (Bepler et al., 2019)). These gains manifest across both high-contrast specimens (Proteasome, PhageMS2) and more challenging low-contrast cases

(Integrin, HumanBAF), suggesting that our model balances sensitivity and specificity more effectively. By selecting a fixed confidence cutoff, we observe a uniform reduction in false positives, directly benefiting downstream reconstruction workflows.

In terms of AUPRC, our method achieves the best results on all four datasets, notably improving over Topaz by +17.8% on PhageMS2 and +10.1% on HumanBAF. While traditional models fail to register meaningful precision at the fixed cutoff, our method maintains valid detection outputs across all test cases, as reflected in the non-zero precision scores. Although all baselines yield near-zero precision under the same conditions, our method preserves confident particle proposals and achieves superior localization accuracy, particularly on Integrin and HumanBAF, where noise and low contrast typically pose significant challenges.

As shown in Figure 8, particle picking accuracy improves significantly when Topaz is fine-tuned using synthetic data generated by CryoCCD. Compared to other data synthesis methods such as Gaussian noise, CycleDiffusion, and CryoGEM, our approach produces more realistic cryo-EM micrographs that better reflect the structural and noise characteristics of experimental data. As a result, the fine-tuned Topaz model yields more accurate particle localization across both static (e.g., HumanBAF) and dynamic (e.g., Integrin, PhageMS2) specimens, closely matching the annotations in real datasets.

## C.2 POSE ESTIMATION

Our approach predicts particle orientations and translations via direct supervised learning on synthetic data. Each training batch contains images with known ground-truth rotations $\{R_i^{\mathrm{gt}}\}$ and translations $\{T_i^{\mathrm{gt}}\}$. We minimize the pose loss:

$$\mathcal{L}_{\mathrm{pose}} = \frac{1}{B}\sum_{i=1}^{B}\left[\frac{1}{9}\|R_i^{\mathrm{gt}} - R_i^{\mathrm{pred}}\|_F^2 + \frac{1}{2}\|T_i^{\mathrm{gt}} - T_i^{\mathrm{pred}}\|_1\right], \tag{15}$$

where $B$ is the batch size, $\|\cdot\|_F$ denotes the Frobenius norm, and $\|\cdot\|_1$ is the $L_1$ norm.

**Evaluation via Reconstruction Resolution (Res(px)).** Predicted poses reconstruct two independent half-volumes using filter back-projection (FBP). Resolution is determined by the Fourier shell correlation (FSC) at threshold 0.5:

$$FSC(r) = \frac{\sum_{r_i \in r} F_1(r_i) \cdot F_2(r_i)^*}{\sqrt{\sum_{r_i \in r}\|F_1(r_i)\|^2 \cdot \sum_{r_i \in r}\|F_2(r_i)\|^2}}, \tag{16}$$

where $F_1$ and $F_2$ are Fourier transforms of the two half-volumes. The resulting Res(px) measures pose accuracy without 2D classification bias.

**Evaluation via Angular Error (Rot(rad)).** We further compute the mean angular error in radians:

$$\mathrm{Rot(rad)} = \frac{180}{\pi\, n_{\mathrm{rots}}}\sum_{i=1}^{n_{\mathrm{rots}}}\arccos\left(\frac{R_i^{\mathrm{gt}}v}{\|R_i^{\mathrm{gt}}v\|} \cdot \frac{R_i^{\mathrm{pred}}v}{\|R_i^{\mathrm{pred}}v\|}\right), \tag{17}$$

using unit vector $v = (0,0,1)$.

**Result Analysis.** Table 3 reports reconstruction resolution (Res(px)) and angular precision (Rot(rad)) for each baseline and our method. Our pose estimator consistently produces sharper reconstructions—reflected in lower Res(px) values—on both static (Proteasome, HumanBAF) and dynamic (Integrin, PhageMS2) specimens, indicating more accurate orientation estimates. Similarly, angular precision improvements demonstrate that a greater proportion of predicted rotations fall within acceptable error bounds, which is key for high-fidelity 3D volume recovery. These overall trends highlight the robustness of our supervised training framework and its capacity to generalize across diverse structural classes.

Compared to CryoFIRE, our method yields consistent gains across all datasets in both resolution and rotation. In particular, we improve Integrin resolution by 55.6% (5.89 vs. 13.27 px) and reduce

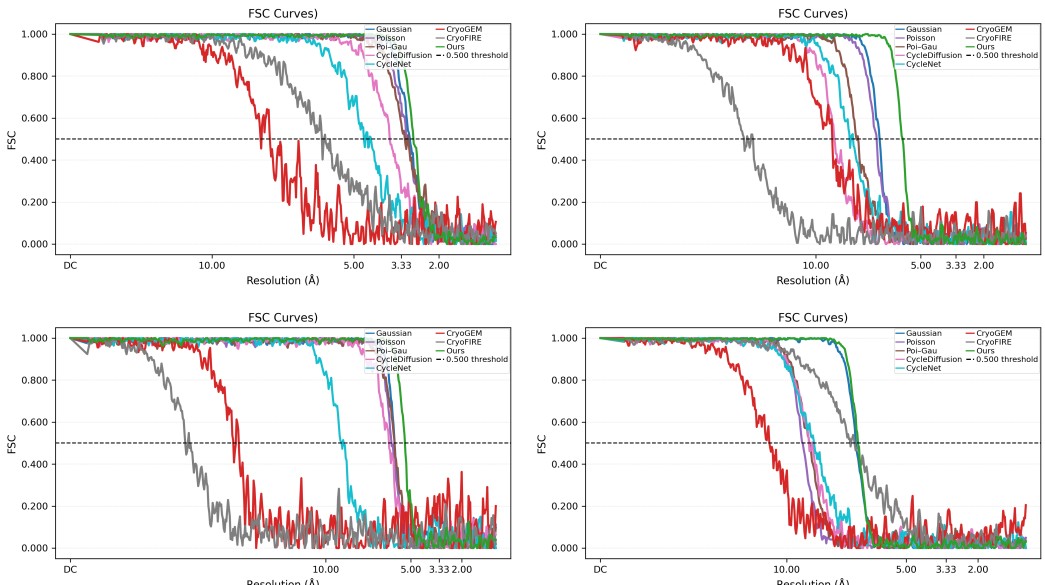

Figure 9: Filter back-projection reconstruction FSC curves at the 0.500 threshold.

PhageMS2 rotation error by 32.0% (0.51 vs. 0.75 rad). We also see large margins on Proteasome—52.3% better resolution (2.87 vs. 6.02 px) and 71.4% lower rotation error (0.44 vs. 1.54 rad)—and a smaller but positive gain on HumanBAF (resolution 7.01 vs. 7.17 px, rotation 1.47 vs. 1.49 rad). On average over the four datasets, our approach reduces resolution error by 45.1% and rotation error by 28.0% relative to CryoFIRE.

Classical noise baselines (Gaussian, Poisson, and their mixture) still underperform across datasets: despite occasionally competitive numbers (e.g., Gaussian resolution on HumanBAF), their average errors remain higher (Res: 5.77–6.55 px; Rot.: 1.14–1.35 rad) than ours (Res: 5.29 px; Rot.: 0.83 rad), reflecting limited noise realism. CryoGEM exhibits the largest errors overall (e.g., Res: 7.99–15.38 px; Rot.: 1.72–1.94 rad), suggesting adversarial distortions that harm both angular accuracy and resolution.

Importantly, our method is best on every dataset and both metrics in Table 3. The margins vary with structural heterogeneity: they are largest on PhageMS2 (complex, flexible; Res: 70.2% better than CryoFIRE) and Proteasome (rigid; Rot.: 71.4% improvement), and narrower on HumanBAF, where we still edge out the next best baseline on both resolution and rotation. These results indicate that our model maintains low errors on both rigid and flexible structures, confirming robustness across structural heterogeneity. Consistent with Figure 9, our advantage in resolution at the 0.500 FSC threshold on Proteasome persists relative to all competing baselines.

# D ABLATION STUDY

## D.1 BIOPHYSICAL MODELING.

Our biophysical engine is implemented as an integrated pipeline; thus only the physics-informed imaging components can be isolated for quantitative ablations. Following this principle, we probe two factors: the *ice layer* model and the *CTF* parameterization. For the ice layer, we replace our physically-based simulation with a global-gradient scheme that randomly selects IceBreaker-estimated thickness maps (Olek et al., 2022). For CTF, we compare (i) **small defocus** (8,000–15,000 Å, 300–800 Å astigmatism), (ii) **large defocus** (10,000–30,000 Å, 500–2,500 Å), and (iii) **ours** (15,000–20,000 Å, 100–500 Å), which targets high-quality cryo-EM imaging. As Table 5 shows, replacing the ice layer with the global-gradient approach severely degrades FID across all datasets, confirming the effectiveness of our ice modeling. Across realistic CTF ranges, FID shifts are within $\leq 7\%$, and our setting is best or on par on five of six datasets (Innexin-6 shows a marginal 0.8% gain under small defocus).

## D.2 Samplers and Sampling Steps.

We validate the effectiveness of different *sampling configurations* on the *Ribosome* dataset. Table 6 presents two complementary studies:

**Sampling steps.** We vary the DDPM step budget from 5 to 50. As expected, the FID steadily decreases as the number of steps increases: from 37.97 (5 steps) to 29.08 (10 steps), 24.44 (20 steps), and finally 20.06 (50 steps). This indicates that longer trajectories allow the denoising process to better approximate the data distribution, although with diminishing returns as the step budget grows.

**Sampler algorithm.** With the step budget fixed to 20, we compare several popular samplers. While DDPM achieves an FID of 24.44, advanced solvers bring consistent improvements: DPM-Solver and DPM-Solver++ reduce FID to 19.27 and 19.13, respectively, while LMS and

Table 6: Quantitative comparison of FID scores on the *Ribosome*: varying DDPM steps vs. different samplers.

| Configuration | Steps | Sampler Algorithm | FID↓ |
|---|---|---|---|
| 5 Steps | 5 | DDPM | 37.97 |
| 10 Steps | 10 | DDPM | 29.08 |
| 20 Steps | 20 | DDPM | 24.44 |
| 50 Steps | 50 | DDPM | 20.06 |
| DDIM | 20 | DDIM | 20.33 |
| DDPM | 20 | DDPM | 24.44 |
| DPM-Solver | 20 | DPM-Solver | 19.27 |
| DPM-Solver++ | 20 | DPM-Solver++ | 19.13 |
| LMS | 20 | LMS | 18.68 |
| UniPC | 20 | UniPC | 18.54 |

UniPC achieve the best performance at 18.68 and 18.54. In contrast, DDIM (20.33) underperforms, likely due to its deterministic formulation, which limits stochastic exploration and reduces sample diversity.

# E Background of Cryo-EM

## E.1 Principles of Cryo-EM

Cryo-EM is a cornerstone of contemporary structural biology, enabling macromolecular structures to be determined at near-atomic resolution in a close-to-native environment (Dubochet et al., 1988; Nogales, 2016). By eliminating the need for crystallization or harsh chemical fixation, cryo-EM allows direct visualization of protein assemblies, viruses, and cellular organelles.

Cryo-EM enables structural imaging of radiation-sensitive biological specimens by operating a transmission electron microscope (TEM) under cryogenic conditions. Electrons, typically accelerated to 300kV, possess a de Broglie wavelength of approximately 0.02Å, vastly shorter than that of visible light (4000–7000Å), thus allowing for atomic-resolution imaging (Henderson, 2015). To mitigate the severe radiation damage caused by electron scattering, cryo-EM employs (i) vitrification via plunge-freezing into liquid ethane to form amorphous ice that preserves the native ultrastructure (Dubochet et al., 1988), and (ii) low-dose imaging strategies, where multiple noisy projections are later computationally averaged.

Modern TEM consists of a high-coherence electron source, condenser lenses that control beam illumination, an objective lens that focuses the electron wavefront after interacting with the specimen, and image magnification lenses. The specimen is maintained below $-170, ^{\circ}$C to prevent devitrification during imaging. Transmitted electrons are collected by a direct electron detector, producing a set of 2D projection micrographs. According to the central projection theorem (DeRosier and Moore, 1970), the Fourier transform of each 2D image corresponds to a central slice of the object's 3D Fourier space. By acquiring thousands of such projections at varying orientations, it becomes possible to reconstruct a 3D structure via algorithms such as iterative back-projection or maximum-likelihood refinement (Cheng, 2018).

## E.2 Major Applications

Single-particle cryo-EM merges tens of thousands of noisy projections into 3D density maps, often reaching 2–3 Å resolution for favourable specimens (Bai et al., 2015). Three computational stages are pivotal:

**Particle picking.** Automatic discrimination of particles from ice, carbon, and contaminants is essential for downstream accuracy.

**Pose estimation.** Assignment of particle orientations is typically achieved via common-line methods or projection matching, both of which are sensitive to noise and heterogeneity.

Demonstrated targets include membrane channels (TRPV1 (Liao et al., 2013; Gao et al., 2016)), soluble enzymes (*E. coli* $\beta$-galactosidase (Bartesaghi et al., 2015)), viral capsids, and chaperonins—each posing unique challenges related to symmetry, flexibility, and low SNR.

## F    USE OF LARGE LANGUAGE MODELS (LLMS)

During the preparation of this paper, we used Generative AI to assist with grammar checking, language polishing, and improving readability. The model was not used for generating novel research ideas, experimental design, data analysis, or drawing conclusions. All content and claims in the paper are the sole responsibility of the authors.

