# OpenReview forum: "CryoCCD: Conditional Cycle-consistent Diffusion with Biophysical Modeling for Cryo-EM Synthesis"
_ICLR.cc/2026/Conference — ICLR 2026 Conference Withdrawn Submission_

### Official Review · Reviewer_3nPm · 2025-10-26

**Soundness:** 3
**Presentation:** 3
**Contribution:** 3
**Rating:** 6
**Confidence:** 4

**Summary:**

The paper presents CRYOCCD, a physics-informed generative framework for synthesizing realistic cryo-electron microscopy (cryo-EM) micrographs. The approach combines (1) a biophysical simulator that generates noise-free projections from 3D molecular structures using ice-thickness modeling and CTF modulation, and (2) a cycle-consistent conditional diffusion model that translates these clean projections into noise-realistic images through mask-guided conditioning, cycle-consistency losses, and a PatchGAN discriminator.
The method is evaluated on several EMPIAR datasets using both visual metrics (FID, CMMD) and downstream tasks—particle picking with Topaz and pose estimation/reconstruction with CryoFIRE + FBP. Results show consistently improved realism and reconstruction quality compared to Gaussian, Poisson, Poisson–Gaussian, CryoGEM, CycleDiffusion, and CycleNet baselines.

**Strengths:**

This paper is well motivated and presents a credible direction toward physics-grounded generative modeling for cryo-EM data. The framework coherently integrates a biophysical simulator with a cycle-consistent conditional diffusion model, showing strong design consistency. Evaluation goes beyond visual fidelity to include practical downstream tasks (particle picking and reconstruction), supported by clear ablations and generalization tests.

**Weaknesses:**

1. No continuous conformational heterogeneity:
The dataset generation pipeline relies on discrete PDB atomic models, without modeling continuous structural variability. As a result, the framework lacks datasets that capture smooth conformational transitions, which are critical for evaluating heterogeneous reconstruction performance.

2. Unspecified feature extractor for FID:
Although FID scores are reported extensively, the paper does not describe the feature extractor used (architecture, training data). Because FID is highly sensitive to the chosen representation, this omission limits interpretability and reproducibility.

3. Unclear mask generation pipeline:
Mask-guided learning is a central component of the proposed method, but the paper does not explain how segmentation masks are obtained—whether they are generated from simulated projections, predicted automatically, or derived from real annotations. Clarifying this is essential to assess scalability to real-world datasets.

4. Narrow reconstruction validation scope:
Reconstruction evaluation is performed only using CryoFIRE combined with Filter Back-Projection (FBP). Including comparisons with other reconstruction frameworks (e.g., CryoDRGN, cryoSPARC refinement) or presenting resolution/error maps would provide a more comprehensive validation.

5. Limited realism under low-SNR conditions:
The Gaussian and Poisson baselines are tested only at SNR = 0.1, which is much higher than typical cryo-EM conditions (SNR ≈ 0.01–0.05). Evaluating the model under lower SNRs would better reflect realistic imaging environments and test the robustness of the simulator.

6. Missing comparisons to recent synthetic datasets:
The paper overlooks several recent works on synthetic cryo-EM data generation and benchmarking, such as CryoBench (Jeon et al.) or CryoPPP (Dhakal et al.). Referencing and comparing against these datasets would strengthen the paper’s positioning within the current literature.

**Questions:**

Already addressed in the 'weakness' section

**Details Of Ethics Concerns:**

No concerns

---

### Official Review · Reviewer_3N8p · 2025-10-27

**Soundness:** 3
**Presentation:** 2
**Contribution:** 2
**Rating:** 4
**Confidence:** 5

**Summary:**

This paper introduces CryoCCD, a synthesis framework designed to generate realistic cryo-electron microscopy (cryo-EM) micrographs. It integrates biophysical modeling with a conditional cycle-consistent diffusion model, which captures the inherent heterogeneity and complex noise characteristics in cryo-EM data. The method employs a biophysical engine that simulates structural diversity and imaging physics, ensuring accurate representation of biological specimens. CryoCCD also incorporates cycle-consistent translation and mask-guided contrastive learning to improve the noise generation process, providing high-fidelity synthetic data. The results show that CryoCCD outperforms existing methods in multiple tasks, including particle picking, pose estimation, and generalization across diverse protein families. The framework demonstrates strong potential to reduce the need for extensive manual annotations and accelerate the development of cryo-EM tools.

**Strengths:**

1. **Originality**: This paper builds upon the foundational work of CryoGEM, but its key innovation lies in the introduction of the CycleDiffusion approach. By combining the power of diffusion models with cycle consistency, the authors present a more robust framework for generating realistic cryo-EM micrographs. The novel integration of diffusion loss and cycle loss enhances the method’s stability and performance, especially in preserving structural fidelity while modeling complex noise. Moreover, the paper expands on the training process by incorporating a larger, more diverse dataset, enabling the model to generalize effectively across mixed datasets, which is a significant advancement over previous generative models.

2. **Quality**: The authors provide sufficient technical details, making the methodology both clear and reproducible. The experimental results are credible and demonstrate realistic simulations, showcasing the method's ability to generate synthetic cryo-EM micrographs.

**Weaknesses:**

1. **Clarity and Methodological Ambiguity**:
Several parts of the paper lack clarity and precise definition, especially in Section 3 and Section 4. The role of the mesh extracted from isosurfaces is never clearly explained—although the paper mentions mesh simplification in the Multi-Scale Volume Modeling stage, it is unclear how or whether the mesh is used in the final projection process. Similarly, the mask generation process is never defined; Section 4 directly introduces masks as inputs without explaining how they are computed or derived. Furthermore, in Section 4, both the notation and logical flow are confusing. After defining two diffusion models, G_AB and G_BA, these are not used in Equations (3)–(6), and several variables such as 𝜖_𝜃 and mask appear without prior definition. The calculation of L_diff lacks a clear description of how G_AB and G_BA participate in the diffusion process. The same issue extends to Equation (9), where it is unclear whether the forward diffusion process involves a single or multiple sampling steps, and the variables x and y are never defined. This lack of formal clarity makes it difficult to follow or reproduce the method.

2. **Conceptual Inconsistency in Motivation (Section 4.2)**:
The paper criticizes GAN-based methods like CryoGEM for not considering cycle consistency, which the authors argue limits their performance. However, the authors still incorporate both contrastive loss and GAN loss—the same losses used in CryoGEM. This creates a conceptual inconsistency: while they criticize CryoGEM for not using cycle consistency, they continue to use the same losses from CryoGEM and do not clearly justify the motivation for their inclusion in the new framework. This redundancy weakens the theoretical coherence of the paper and calls for a clearer explanation of why these losses are necessary in the proposed method.

3. **Weak Baseline Comparison**:
The evaluation of CryoGEM as a baseline is highly confusing. The reported CryoGEM results are far worse than those in the original paper, even though the CryoGEM code, pretrained weights, and datasets are publicly available (e.g., EMPIAR-10028). For instance, in Table 2, the AUPRC for particle picking on PhageMS2 drops from 0.915 (in the original paper) to only 0.159, and in Figure 4, the Poisson noise simulations fail completely. This raises doubts about whether CryoGEM was correctly reproduced or appropriately configured. One possibility is that CryoGEM, which was designed as a per-scene training method, may have been trained on mixed datasets (15 different scenes), which is inconsistent with its intended use. The paper should provide a clear justification for why the baseline fails to converge and, for fairness, compare CryoCCD against CryoGEM’s official pretrained models or datasets.

4. **Experimental Limitations and Generalization Concerns**: Although CryoCCD claims to generalize across heterogeneous data, the experimental results suggest otherwise. In Figure 4, the generated synthetic micrographs show substantial visual differences from real data, and the FID scores consistently exceed 100, indicating limited realism. Conceptually, it is also unclear how a diffusion model can learn to generate realistic micrographs without conditioning on microscope parameters such as defocus or dose rate. Since each biological structure in the dataset corresponds to only one specific noise distribution, training a single diffusion model to generalize across multiple noise conditions seems questionable. Without additional conditioning variables, the model lacks the information needed to decide what type of noise to produce, which may explain the poor quantitative metrics and unstable training dynamics. The paper would benefit from ablation studies or additional experiments clarifying how CryoCCD handles this inherent heterogeneity.

**Questions:**

The direction of this paper is promising, and I appreciate the substantial engineering effort and technical details presented. However, the weaknesses outlined above are too significant for me to confidently recommend acceptance at this stage. I encourage the authors to address these issues to substantially strengthen the paper. Thanks!

---

### Official Review · Reviewer_ggqZ · 2025-11-01

**Soundness:** 3
**Presentation:** 2
**Contribution:** 2
**Rating:** 2
**Confidence:** 5

**Summary:**

The paper presented CryoCCD, a framework/method that uses conditional cycle-consistent diffusion to generate realistic cryo-EM micrographs with given particles and imaging parameters. The authors evaluated the results with other baselines with several metrics and also demonstrated that CryoCCD can generate synthetic micrographs that benefit some downstream tasks.

**Strengths:**

- To my knowledge, this is the first diffusion framework to learn the noise generation in cryo-EM. Noise model in cryo-EM is crucial the the downstream reconstruction but also very difficult to model due to the complexity of the experimental data.

- The results, including the visual quality, common CV metrics, and the metrics for particle picking and pose accuracy are quite impressive compared to the baselines.

**Weaknesses:**

- Synthesizing realistic cryo-EM micrographs is an interesting task, especially the biggest challenges in the computational cryo-EM include the lack of ground truth and the gap between real and synthetic data. However, in my opinion, the goal of this task is not actually generating "real-looking" micrographs, but to demonstrate how does better synthetic data benefit real problems in cryo-EM data processing. The authors indeed show the benefit of CryoCCD in particle picking and pose estimation. However, in particle picking, Topaz is a method published over 6 years ago and there are a lot of better particle pickers that can be finetuned in the same way. CryoFIRE is not the mainstream ab initio modeling method and in practice, the most robust pose estimation methods still use searching instead of gradient descent based inference. These undermines the usefulness of how a better synthetic method can benefit real processing.

- Related to the previous step, another problem in both downstream applications are the need of a ground truth reconstruction to perform the physics-based simulation. However, the overall goal of the cryo-EM processing pipeline is high quality reconstruction.

- As of the generalization ability (Sec. 5.4), in the held out datasets (I would not call it "protein families"), both TRPV1 and ribosome are similar to the particles in the training set. EMPIAR-10389 is a highly symmetrized globular shaped complex which also resembles a few samples in the training set. Additionally, the authors only compared the FID and CMMD scores with the baselines.

- Limited novelty: although this is probably the first application of using diffusion models to synthesize cryo-EM noise, the methods themselves lack novelty. The "biophysical engine" part seems interesting, but many elements, e.g. the conversion from atomic models to densities, the projection model at different poses is widely known and applied.

**Questions:**

1. In Fig 1(a), what do "processed particles" mean here? Does it refer to just the processed 3D density maps, or does it refer to the processing results of reconstructed particles (2D projections)?

2. Why use a triangulated mesh to represent the densities (L157)? Why not use the typical voxel grid?

3. For particle coordinates and poses, the authors use "RELION-derived picks and angles". Where does the this information come from? In class specific distribution sampling, where do the "experimental distributions" come from? Are they empirical summary, or learned from a large scale of data?

4. What does "multi-scale volume modeling" mean? Is this to simulate different magnification of the microscope? Please elaborate. Also "ensures consistent biological realism from whole-cell panoramas down to molecular interfaces" seems to be an overclaim since cellular environment is much more than simulating projections of biomolecules.

5. In ice layer modeling, what's the physical insight behind the Perlin-noise? Additionally, what is "beam-induced noise" in the absence of any particles?

6. What is rho_i in Eq.1?

7. How is the ice layer modeling connecting to the projection in Eq.2?

8. I don't believe that cryoFIRE is the "state-of-the-art ab-initio reconstruction methods". I understand for the purpose of the experiment, the authors want to show that using synthetic data to train a pose estimation model can benefit the result, but in practice, for ab initio reconstruction, people still heavily use cryoSPARC's method.

9. How is the structure library connected to the overall experiments, since only a few selected proteins/complexes are used to generate the synthetic micrigraphs?

10. In the appendix, EMPIAR-10363 and EMPIAR-10421 are not properly introduced.

11. When is CTF applied to the synthesize pipeline? How are the defocus values determined?

12. Can the authors also compare the synthetic micrographs (including baselines) vs real in the Fourier space, by the power spectrum?

---

### Official Review · Reviewer_rSVn · 2025-11-02

**Soundness:** 2
**Presentation:** 2
**Contribution:** 2
**Rating:** 2
**Confidence:** 4

**Summary:**

The paper proposes a data generation framework aimed at improving the training of cryo-EM reconstruction pipelines. The method leverages a large collection of 3D molecular structures from PDB and AlphaFold3, combined with biophysical priors in the image formation process, to simulate realistic cryo-EM images. A diffusion-based generator is then trained to synthesize images that better capture the noise characteristics of experimental data. The authors evaluate the generated data both in terms of image realism and its utility for downstream cryo-EM tasks, including particle picking, pose estimation, and 3D reconstruction, comparing against several prior data-generation methods.

**Strengths:**

- The proposed strategy is reasonable for improving data-driven cryo-EM analysis.

- The paper includes comparisons with multiple baseline methods, demonstrating effort toward comprehensive evaluation.

**Weaknesses:**

- Incremental novelty. The overall synthetic data generation framework closely follows CryoGEM, with the primary differences being the use of a larger molecular library and the replacement of the GAN generator with a standard diffusion model (partially). Both modifications seem to be straightforward extensions.

- Annotation and applicability limitations. The proposed simulation pipeline requires annotations of molecular orientations and particle-type-specific priors, increasing annotation cost and potentially limiting the applicability of the approach. Additionally, the motivation for multi-scale volume modeling is unclear, as cryo-EM images typically do not involve “whole-cell panoramas”.

- Complex and loosely integrated model design. The training framework combines diffusion modeling with elements of GAN-based objectives, resulting in a hybrid loss (Eq. 12) composed of multiple terms from prior works without a clearly unified formulation. This design may complicate optimization and hyperparameter tuning, and the paper does not sufficiently justify the loss design conceptually or via ablation study.

- Unconvincing experimental comparisons. The reported performance of CryoGEM baselines differs substantially from those in Zhang et al., NeurIPS 2024, raising concerns about the fairness and reproducibility of comparisons. Specifically:
    (1) The qualitative examples of CryoGEM in Figure 4 differ significantly from those in the original paper.
    (2) The FID and FSC scores are markedly worse than the reported values in CryoGEM.
    (3) The performance on particle picking and pose estimation is also notably lower than in CryoGEM.

    Additionally, while the paper claims that GAN-based methods suffer from mode collapse, no empirical evidence is provided. The ablation study is also limited and would benefit from a more detailed analysis of individual components of the proposed biophysical modeling pipeline.

**Questions:**

- Please clarify the additional annotation requirements and discuss their implications for scalability and practical deployment.

- Could you reconcile the experimental discrepancies with prior CryoGEM results (e.g., Zhang et al., NeurIPS 2024) and explain the potential causes?

---

### Note · Authors · 2025-11-12

**Comment:**

We sincerely thank all reviewers and area chairs for their valuable time and constructive feedback. We will carefully consider their comments in our future revision.

**Withdrawal Confirmation:**

I have read and agree with the venue's withdrawal policy on behalf of myself and my co-authors.